# Development of Bioactive Hybrid Poly(lactic acid)/Poly(methyl methacrylate) (PLA/PMMA) Electrospun Fibers Functionalized with Bioglass Nanoparticles for Bone Tissue Engineering Applications

**DOI:** 10.3390/ijms25136843

**Published:** 2024-06-21

**Authors:** Fabián Álvarez-Carrasco, Pablo Varela, Mauricio A. Sarabia-Vallejos, Claudio García-Herrera, Marcela Saavedra, Paula A. Zapata, Diana Zárate-Triviño, Juan José Martínez, Daniel A. Canales

**Affiliations:** 1Laboratorio de Biomecánica y Biomateriales, Departamento de Ingeniería Mecánica, Facultad de Ingeniería, Universidad de Santiago de Chile, Santiago 9160000, Chile; fabian.alvarez.c@usach.cl (F.Á.-C.); pablorafaelvv@gmail.com (P.V.); 2Facultad de Ingeniería, Arquitectura y Diseño, Universidad San Sebastián, Santiago 8420524, Chile; mauricio.sarabia@uss.cl; 3Grupo Polímeros, Departamento de Ciencias del Ambiente, Facultad de Química y Biología, Universidad de Santiago de Chile, Casilla 40, Correo 33, Santiago 9160000, Chile; marcela.saavedra@usach.cl (M.S.); paula.zapata@usach.cl (P.A.Z.); 4Laboratorio de Inmunología y Virología, Facultad de Ciencias Biológicas, Universidad Autónoma de Nuevo León, San Nicolás de los Garza 66455, Mexico; diana.zaratetr@uanl.edu.mx; 5Centro de Ingeniería y Desarrollo Industrial, Av. Playa Pie de la Cuesta No. 702, Desarrollo San Pablo, Santiago de Querétaro 76125, Mexico; juanjose.martinez@cidesi.edu.mx; 6Instituto de Ciencias Naturales, Facultad de Medicina Veterinaria y Agronomía, Universidad de Las Américas, Manuel Montt 948, Santiago 7500975, Chile

**Keywords:** PLA/PMMA-based materials, bioglass nanoparticles, bioactive scaffolds, bone tissue engineering

## Abstract

Hybrid scaffolds that are based on PLA and PLA/PMMA with 75/25, 50/50, and 25/75 weight ratios and functionalized with 10 wt.% of bioglass nanoparticles (n-BG) were developed using an electrospinning technique with a chloroform/dimethylformamide mixture in a 9:1 ratio for bone tissue engineering applications. Neat PLA and PLA/PMMA hybrid scaffolds were developed successfully through a (CF/DMF) solvent system, obtaining a random fiber deposition that generated a porous structure with pore interconnectivity. However, with the solvent system used, it was not possible to generate fibers in the case of the neat PMMA sample. With the increase in the amount of PMMA in PLA/PMMA ratios, the fiber diameter of hybrid scaffolds decreases, and the defects (beads) in the fiber structure increase; these beads are associated with a nanoparticle agglomeration, that could be related to a low interaction between n-BG and the polymer matrix. The Young’s modulus of PLA/PMMA/n-BG decreases by 34 and 80%, indicating more flexible behavior compared to neat PLA. The PLA/PMMA/n-BG scaffolds showed a bioactive property related to the presence of hydroxyapatite crystals in the fiber surface after 28 days of immersion in a Simulated Body Fluids solution (SBF). In addition, the hydrolytic degradation process of PLA/PMMA/n-BG, analyzed after 35 days of immersion in a phosphate-buffered saline solution (PBS), was less than that of the pure PLA. The in vitro analysis using an HBOF-1.19 cell line indicated that the PLA/PMMA/n-BG scaffold showed good cell viability and was able to promote cell proliferation after 7 days. On the other hand, the in vivo biocompatibility evaluated via a subdermal model in BALC male mice corroborated the good behavior of the scaffolds in avoiding the generation of a cytotoxic effect and being able to enhance the healing process, suggesting that the materials are suitable for potential applications in tissue engineering.

## 1. Introduction

Tissue engineering (TE) is an emerging, multidisciplinary field that combines different areas, such as medicine, biology, and materials science, among others, to provide materials that can cure, replace, or regenerate any tissue, organ, or human body function [1,2,3]. Bone regeneration is mainly associated with using a support material, called scaffolding, and cells or bioactive molecules that, together, guide and promote the in situ regenerative process of the affected area [4,5]. Scaffolds that possess a fiber structure, high porosity, and pore interconnectivity obtained via the electrospinning technique have been extensively investigated for these applications [6,7,8].

Electrospinning is a simple and versatile technique that makes obtaining micro- or nanometric-sized fibers possible. For bone regeneration, it is crucial since they can imitate the extracellular matrix (ECM), favoring the cellular processes necessary for this purpose [9,10]. Electrospun fibers based on biocompatible and biodegradable polymers, such as poly (lactic acid) PLA, have been developed as potential scaffolds in tissue engineering, particularly in terms of their capability for cell regeneration processes [11,12,13]. PLA is a synthetic thermoplastic biopolymer with a wide field of applications. Interest in this material in biomedicine and tissue engineering is due to its excellent biocompatibility and biodegradability, as well as its mechanical properties, similar to those of olefin-derived polymers. It is non-toxic, which enables its potential application in the human body [14,15]. Despite having good advantages over many materials, PLA has properties that limit its applications in some areas related to its high rigidity, low impact resistance, low hydrophilicity, cell affinity, and lack of bioactivity under physiological conditions [16,17]. Given these limitations, combining PLA with other polymers is considered a novel method that allows for obtaining hybrid materials with improved properties that broaden their range of use, such as in bone regeneration therapies [18,19,20]. Acrylic polymers such as poly (methyl methacrylate) PMMA have been widely used in biomedicine due to their good compatibility, chemical resistance, low toxicity, good hydrophilicity, and better toughness than matrices such as PLA. In addition, they have been approved by the Food and Drug Administration (FDA) for use in fields such as ophthalmology, dentistry, and tissue engineering [21,22,23]. PMMA, in combination with other matrices, has been reported to obtain hybrid matrices with improved properties. In this sense, Anakabe et al. prepared a mixture of Poly (lactic acid) (PLA) with poly (methyl methacrylate) (PMMA) via melt mixing, using different PLA/PMMA ratios, (80/20), (60/40), (50/50), (40/40), and (20/80). Their results regarding mechanical characterization indicate that the incorporation of PMMA decreased the elastic modulus of the mixtures compared to PLA, showing a more flexible character. Furthermore, this behavior became more noticeable with an increase in PMMA in the solution [24]. In addition, Son et al. reported developing PCL/PMMA fibrous scaffolds for bone tissue engineering applications using an electrospinning technique; they prepared a polymer blend with 10/0, 7/3, 5/5, and 3/7 weight ratios. Their results indicated that adding PMMA increased the fiber diameter and improved the wettability of the scaffolds. Along with this, the authors suggest that the PCL/PMMA scaffold (7/3 ratio) was adequate for cell growth in vitro using an MG-63 cell line; these results were corroborated through in vivo studies showing new tissue formation, suggesting that this compound is a potential scaffold for bone regeneration [25]. Despite the significant improvement that the incorporation of PMMA could provide to PLA matrices, both types of materials lack bioactivity, which, in bone tissue engineering, is the ability of a material to promote mineralization or the formation of a layer of hydroxyapatite (HA) under physiological conditions, favoring the regenerative process [26,27]. The functionalization of polymer matrices such as PLA, incorporating bioactive nanoparticles such as bioactive glass (n-BG), has been widely reported [28,29,30,31,32,33]. Bioactive glasses are ceramic materials based on SiO_2_-CaO-P_2_O_5_ ternary systems with excellent biocompatibility and a strong bioactive character, promoting a strong bond between the material and the tissue when in contact with biological fluids [34,35,36]. Liverani et al. reported the development of electrospun scaffolds based on PCL/Chitosan with the incorporation of 30% n-BG 4S5S bioactive glass nanoparticles (20–80 nm). The scaffolds showed the ability to promote the growth of hydroxyapatite (HA) in SBF solution after 7 days of immersion in SBF [37]. Noh et al. prepared electrospun PLA fiber mats by incorporating a 10 wt.% of BG (70% SiO_2_–25% CaO–5% P_2_O_5_), obtained via a fragmentation process and with an average diameter size of 182 nm. Bioactivity analysis showed HA mineralization after 7 days of immersion in an SBF solution. In addition, the nanocomposites showed good cell adhesion, viability, and proliferation when pre-osteoblastic cells were used [33]. Canales et al. prepared PLLA-based electrospun scaffolds incorporating 10 and 20% n-BG (23 nm). The scaffolds showed a strong bioactive character after 14 days of immersion in SBF. In addition, they presented good cell viability and differentiation compared to pure PLA using cells derived from bone marrow from line ST-2 [16]. Rong et al. prepared a fiber scaffold for bone tissue engineering based on PLA/PMMA blend with a 1:1 relation and with the incorporation of 4 wt.% of nano-hydroxyapatite (n-HA) with an average size of 20 nm as a bioactive filler for bone tissue engineering applications; their results indicated that the PLA/PMMA/n-HA has good spinnability, obtaining a porous structure with pore interconnectivity. The bioactive analysis under an SBF solution after 14 days showed that the PLA/PMMA/n-HA promoted the growth of hydroxyapatite crystals, and the in vitro biological characterization using an MG-63 cell line revealed that the scaffolds showed good cell viability, being an interesting matrix for these kind of applications [38]. These results showed that the incorporation of nanoparticles into polymeric matrices is an effective route for the development of functional scaffolds with applications in bone regeneration; however, the development of scaffolds based on PLA/PMMA mixtures reinforced with bioactive glass (n-BG) has not been reported.

Considering these previous reports, in this work, hybrid electrospun matrices based on poly (lactic acid) and poly (methyl methacrylate) PLA/PMMA were developed in 75/25, 50/50, and 25/75 ratios and functionalized with a 10 wt.% of bioactive glass nanoparticles as a potential scaffold for applications in bone tissue engineering. Morphology, fiber size, mechanical properties, wettability, hydrolytic degradation, and bioactivity were determined and compared with those of pure PLA. In addition, cell viability was evaluated through in vitro assays using a human fetal HFOB-1.19 cell line. In parallel, biocompatibility was determined through in vivo analysis using a dorsal subdermal implantation in male BALC mice.

## 2. Results and Discussion

### 2.1. Characterization of Bioglass Nanoparticle (n-BG)

Bioactive glass nanoparticles (n-BG) were morphologically characterized via Transmission Electron Microscopy (TEM). Figure 1A shows that the nanoparticles have a spherical morphology, are highly agglomerated and have an average size of 23 ± 4 [nm]. Figure 1B shows a size histogram, revealing a unimodal distribution between 18 and 24 nm. Figure 1C,D show the Infrared Spectroscopy (FT-IR) and X-ray Diffraction (XRD) analysis, respectively. The FT-IR spectra show the characteristic signals for this kind of nanoparticles, a peak close to 1072 cm^−1^ and 803 cm^−1^ associated with the vibrations of the asymmetric and symmetric stretching of the Si–O–Si bonds, respectively. Similarly, a peak obtained around 467 cm^−1^ was mainly related to strain vibrations of Si–O–Si and O–Si–O bonds. Finally, the last two signals found, one around 950 cm^−1^ characteristic of Si–O–Ca bond vibrations, was superimposed on the Si–O–Si bonds at 1072 cm^−1^ and confirm the presence of calcium in the synthesized nanoparticles and a peak close to 578 cm^−1^, attributed to the deformation vibrations of the P–O bonds related to vibrations of the carbonate ion (CO_3_^−2^) derived from the dissolution of CO_2_ from the atmosphere during sol-gel synthesis [16,30,35,39].

For the XRD pattern, it was observed that the nanoparticles present an amorphous structure, as previously reported. Furthermore, the presence of a broad and low-intensity band that goes from 15 to 35° characteristic of the amorphous SiO_2_ network for this type of nanoparticle, confirms n-BG’s nature [30].

### 2.2. Morphology and Size of Hybrid Electrospun Fibers

Figure 2 shows the morphology of the PLA- and PMMA-based scaffold in the different proportions prepared, analyzed via Scanning Electron Microscopy (SEM), and the images were processed using the ImageJ software 1.53. The fibers based on pure PLA presented a random and uniform structure with high porosity and pore interconnectivity. The PLA/PMMA hybrid fibers showed a similar random arrangement; however, as the PMMA content in the filler increased, the structure became less uniform, and they presented defects (beads). For sample PP02, a structure very similar to that of PLA was observed, being homogeneous and without the presence of beads. For PP03, a porous structure was observed, but it was more irregular and showed the presence of defects (beads). After increasing the amount of PMMA in the PP04 composition, a greater number of beads with a considerably larger size could be seen, affecting the homogeneity of the structure. Finally, for the PMMA-based scaffold, no fiber formation was observed, indicating that the solvent system used (CF/DMF; 9:1) disrupted the electrospinning process. Similar results were reported by Lv et al. They prepared nanofibers based on combinations of stereo complex-Poly (lactic acid) and Poly (methyl methacrylate) sc-PLA/PMMA in relations of 0/100, 10/90, 15/85, 20/80, and 25/75. When the amount of sc-PLA increased, the structure obtained presented better homogeneity and porosity; on the contrary, the structure based on PMMA presented beads and was more irregular [20]. For the PLA/PMMA scaffolds with the incorporation of 10 wt.% of bioglass nanoparticles (n-BG), the SEM images show similar behavior; when the PMMA content was higher, the structure of the developed fiber lost homogeneity, and it was possible to observe the presence of beads. Furthermore, PPBG01 showed beads in the structure, and these defects can be attributed to nanoparticle agglomerations. All PLA/PMMA/n-BG hybrid scaffolds also showed these defects in the structure. In previous works, Canales et al. reported electrospun fibers’ preparation based on PLA incorporating bioglass nanoparticles with 10 and 20 wt.%. The morphological results indicate that the incorporation of n-BG promotes the appearance of defects, and these increase with the content of nanoparticles. This phenomenon is attributed to the low matrix–nanoparticle interaction that induces their agglomeration on the surface of the fibers [16].

Figure 3A shows the results of the statistical analysis for the fiber diameters of the developed scaffolds; the PP01 had an average size of 0.93 ± 0.32 µm, and for the PP02 scaffold, the value was 1.28 ± 0.40 µm, showing a significant increase (*** = *p* < 0.001) with respect to PLA. Meanwhile, the PPBG03 and PPBH04 scaffolds obtained a value of 0.92 ± 0.37 and 0.70 ± 0.28 µm, respectively, the latter being significant compared to PLA. Our results suggest that the incorporation of PMMA causes a decrease in fiber size; this happened when the concentration of PMMA was equal to or greater than that of PLA. Lv et al. showed a similar effect when they prepared nanofibers based on sc-PLA/PMMA in ratios of 0/100, 10/90, 15/85, 20/80, and 25/75. When the PLA content increased, a greater average fiber diameter was observed in all the relationships studied [20].

A similar pattern was observed for the PLA and PLA/PMMA scaffolds functionalized with n-BG; with a higher PMMA content, the fiber size decreased. The values obtained for PPBG02, PPBG03, and PPBG04 were 1.82 ± 0.98, 1.32 ± 1.01, 0.72 ± 0.59, and 0.33 ± 0.09 µm, respectively. In addition, it was observed that for the PPBG01 scaffold, the average diameter increased by about 95% compared to pure PLA; this increase was mainly associated with the effects that the incorporation of n-BG caused on the conductivity of the solution affecting the electrospinning process [16,31]. However, this increase was not observed for the PLA/PMMA/n-BG mixtures; on the contrary, the values decreased compared to the combinations without nanoparticles. These results suggest that, in PLA/PMMA/n-BG blends, the average fiber size is preponderantly affected due to the presence of PMMA compared to the presence of n-BG.

### 2.3. Wettability of Hybrid Electrospun Fibers

The wettability of a material, determined via contact angle (CA) measurements, is an important parameter that provides information about the hydrophobic or hydrophilic nature of a biomaterial [40]. The polar character of the sample is essential for applications such as tissue engineering since it greatly influences the material–cell interaction, impacting the biological behavior of the material under physiological conditions [41]. Depending on the contact angle value, materials can be classified as hydrophilic with a CA of less than 90°, hydrophobic if the CA values range from 90 to 120°, and super hydrophobic with a CA above 120° [40].

Figure 3B shows the contact angle results obtained for the PLA and PLA/PMMA scaffolds. PP01 showed a contact angle value of 99.96 ± 4.33°, confirming the hydrophobic nature of this material [40]. The contact angle values for the PP02 and PP03 mixtures decreased to 95.03 ± 1.12 and 89.64 ± 1.01°, respectively. Although our results indicate that the incorporation of PMMA reduces the contact angle value, in statistical terms, this variation was not significant. A similar behavior was reported by Son et al. who prepared electrospun scaffolds based on PCL/PMMA mixtures in 70/30, 50/50, and 30/70 ratios, finding that the incorporation of PMMA decreased the CA value of the PCL from 110 ± 0.3° to 95 ± 0.4 for the PCL/PMMA 30/70 scaffold. This phenomenon is mainly due to the hydrophilic nature of PMMA, resulting in a decrease in the CA value as its content in the scaffolds increases. Previous studies have shown the hydrophilic nature of PMMA, presenting a contact angle value close to 67.8 ± 1.4 [24].

For the scaffolds functionalized with BG nanoparticles, the contact angle values were 101.23 ± 1.86°, 97.82 ± 2.12°, and 93.96 ± 0.77° for PPBG01, PPBG02, and PPBG03, respectively. The behavior is similar to that of the scaffolds without an n-BG presence; as the PMMA content increases, the CA value decreases. However, although the presence of n-BG also did not modify the hydrophobic behavior of the scaffolds, it is possible to observe that the CA values of scaffolds are slightly higher with n-BG. In general, the higher hydrophobicity of randomly electrospun materials, regardless of the type of polymer used and its intrinsic characteristics, is mainly related to high porosity, the area/volume ratio, the specific surface area, and a fine pore structure that increases their hydrophobic character [42].

### 2.4. Infrared Spectroscopy Analysis of the Hybrid Electrospun Fibers

Figure 4A presents the FT-IR spectra obtained from the examination of the electrospun fibers. For the pure PLA scaffold, characteristic signals were found around 1752 and 1186–1081 cm^−1^, associated with the stretching vibration of the C=O bonds and the stretching vibration of the C–O bonds related to the acid group, respectively. In addition, another signal located near 1360 cm^−1^ corresponds to the C–C bond strain vibration, with a peak close to 1453 cm^−1^, which was related to the C–H bond strain vibration of the methyl group [17,19]. For the PMMA-based scaffold, signals were found at around 1724 and 1148–1245 cm^−1^ associated with stretching vibration of the C=O bonds and stretching vibration of the C–O bonds of the ester group, respectively [43]. In addition, signals observed at 1140 and 900 cm^−1^ were related to deformation vibrations of C–C and C–H bonds of the methyl groups. Finally, a double band was found at 2952 and 2995 cm^−1^, characteristic of stretching vibrations of the C–H bonds of the methyl group [22]. In the case of the PLA/PMMA scaffolds in different ratios, the characteristic bands for PLA and PMMA can be observed. However, they are superimposed, indicating no changes in the chemical structure of either polymer in the mixture.

The n-BG incorporation can be seen in Figure 4B; all spectra show a peak around 467 cm^−1^ attributed to the characteristic strain vibrations of Si–O–Si and O–Si–O bonds [29].

### 2.5. Mechanical Characterization of Hybrid Fibers

Analyzing mechanical behavior is very important when evaluating the potential of a material for applications in bone regeneration. The material must support the tissue’s biomechanical requirement to be replaced, which, in turn, must be stable during the regenerative process [44]. The mechanical properties of the PLA and PMMA hybrid scaffolds were evaluated in terms of Young’s modulus, tensile strength, and the elongation at break through uniaxial stress tests to analyze the effect of the incorporation of PMMA, as well as the incorporation of n-BG in the mechanical behavior of PLA.

Figure 5A shows the typical stress–strain curve, where it is possible to observe the elastic and plastic deformation zones, and Figure 5B–D show the statistical analysis results of mechanical properties for prepared scaffolds. Regarding Young’s modulus, in general, with the incorporation of PMMA, a decrease in values can be seen; while the PP01 scaffold obtained a value of 10.87 ± 4.74 MPa, the PP02 and PP03 scaffolds obtained values of 8.95 ± 3.62 and 6.60 ± 1.88 MPa, respectively, the latter being statistically significant compared to PLA alone (* = *p* < 0.05). Regarding the tensile strength, a significant decrease was observed compared to pure PLA, reaching values of 0.614 ± 0.085, 0.317 ± 0.074, and 0.399 ± 0.044 MPa for PP01, PP02, and PP03, respectively. These results may have been obtained since PMMA has a more ductile behavior than PLA, decreasing the parameters studied [20,24]. On the other hand, the results obtained for the elongation at break showed a different effect; the PLA scaffold had a value of 115.08 ± 20.43%, and the PP02 hybrid scaffold showed an increase, reaching a value of 142.74 ± 10.25%; finally, for the PP03 scaffold, a decrease was observed, reaching a value of 74.53 ± 14%. These results indicate that, for the PP02 combination, an increase in elongation is consistent with the decrease in modulus and resistance. A more ductile behavior was perceived in comparison to pure PLA. However, this effect changed for the PP02 scaffold; a decrease in elongation was observed, which was mainly due to the greater presence of defects in the structure, which act as fracture points, decreasing the elongation capacity of the scaffold [45]. Canales et al. reported the mechanical characterization of PLA-based electrospun scaffolds with the addition of different nanoparticles, finding similar values (11.1 ± 1.8 MPa, 0.20 ± 0.05 MPa, and 113.7 ± 21.3%) for Young’s modulus, tensile strength, and elongation at break, respectively, of PLA fibers [16]. Son et al. reported the mechanical properties of the effect of PCL/PMMA electrospun fibers; their results indicated that the addition of PMMA decreased the mechanical behavior concerning pure PCL, and they associated these phenomena mainly with the increase in the fiber diameter with the increase in the PMMA ratio in the blend [25]. On the other hand, the effect of nanoparticles on the mechanical behavior was also analyzed. The PPBG01 scaffold presented a significant decrease in the three parameters evaluated compared to PP01, obtaining a value of 6.22 ± 1.44 MPa, 0.378 ± 0.070 MPa, and 51.19 ± 4.90% for Young’s modulus, tensile strength, and the elongation at break, respectively. For the PLA/PMMA/n-BG scaffolds in the relations studied, a decrease was also observed concerning PLA, reaching values of 7.05 ± 1.87 MPa, 0.562 ± 0.091 MPa, and 53.86 ± 12.96% for the PPBG02 and 2.28 ± 1.08 MPa, 0.214 ± 0.050 MPa, and 50.31 ± 11.73% for PPBG03 in Young’s modulus, tensile strength, and the elongation to breakage, respectively. Liverani et al. reported similar results when incorporating 30% n-BG into mixtures of PCL and chitosan. They found that adding n-BG significantly decreased the mechanical properties due to the increase in the diameter of the fibers and the appearance of defects in the scaffolds [37]. Currently, the effect of many parameters on the mechanical properties of electrospun scaffolds has been studied [46,47,48]. Alharbi et al. studied the effect of molecular weight and fiber diameter on the mechanical properties of PCL scaffolds. They obtained results showing that the elastic modulus decreased drastically with the increase in fiber diameter, while there were no significant variations with the change in molecular weights of PCL [49]. While Rashid et al. concluded that the appearance of defects such as beads in fibers alters their mechanical properties by creating stress concentration points and structural discontinuities, leading to premature fracture and unequal load distribution [45]. These defects also generate weak interfaces with the polymer matrix, reducing tensile strength and elasticity and, in general decreasing the load capacity and deformability of the fibers [50,51]. Similarly, in our study, scaffolds that included PMMA and n-BG showed larger fiber diameters and the appearance of imperfections, these two factors being the main reasons related to the decrease in mechanical properties.

### 2.6. In Vitro Bioactivity

The bioactive properties of PLA and PMMA scaffolds functionalized with 10 wt.% of n-BG were evaluated after 7, 14, 21, and 28 days of immersion in a Simulated Body Fluid solution (SBF). Figure 6 shows the SEM images of scaffolds after 21 days, and it is possible to observe that the samples without bioglass do not have the presence of a hydroxyapatite (HA) layer on the surface of fibers. However, after bioglass incorporation, the samples are capable of promoting HA mineralization. The HA crystals are spherical, and they can be appreciable on all the surfaces of fibers. HA mineralization is essential in bone tissue engineering to generate an adequate chemical environment to induce the necessary biological processes for bone regeneration [26]. The bioactive properties of bioglass nanoparticles and their use as a functional biomaterial to reinforce polymer matrices are reported in previous works as a promissory biomaterial for bone regeneration therapies [35,52,53]. Canales et al. reported the preparation of electrospun scaffolds based on PLA nanocomposites with the incorporation of mixtures of nanoparticles of bioglass and magnesium oxide, with a size of 25 and 23 nm, respectively, and a (5/5) and (10/10) weight relation, to obtained bioactive and antimicrobial bifunctional porous scaffolds. Their results indicated that the PLA/n-BG/n-MgO showed bioactivity after 14 days, and on the other hand, they showed an antimicrobial effect against *Staphylococcus aureus* after 6 h of exposure. The MgO nanoparticles did not show bioactive properties, and the bioactive ability was attributed to n-BG filler [16].

The formation of a hydroxyapatite layer on the fiber surface was confirmed via SEM images and corroborated via SEM-EDS analysis. Table 1 presents the Ca/P ratio obtained for the samples studied via EDS analysis. The scaffolds without n-BG did not show the presence of calcium (Ca) or phosphorous (P), but for PPBG01, the Ca/P relation reached a value of 1.90, and for PPBG02 and PPBG03, the values of Ca/P were 2.06 and 1.87, respectively. The values obtained are close to 1.67, which is the Ca/P value for bone-native hydroxyapatite [30].

### 2.7. In Vitro Hydrolytic Degradation

Due to its support function, the structural stability of the scaffolds under physiological conditions is an essential parameter to evaluate for tissue engineering applications. In this sense, the degradation rate of prepared scaffolds was studied at different immersion times, including 3, 7, 14, 21, 28, and 35 days in a phosphate-buffered saline solution (PBS). The degradation process was analyzed via SEM, the mass loss percentage, and pH measurements. Figure 7A shows the SEM images of the samples after 0 (left) and 35 (right) days of study. These results revealed that the samples with n-BG degraded faster compared to the samples without nanoparticles. In terms of mass loss (Figure 7B), it was observed that the PP01 scaffold experienced a 9% loss, while the PP02 and PP03 samples lost 4% and 3%, respectively. These results indicate that the inclusion of PMMA reduces the degradation rate of pure PLA.

Rong et al. reported a similar behavior when they prepared electrospun scaffolds based on PP03. The degradation rate of the mixture was lower than that of PP01, which was mainly attributed to the lower degradation rate of PMMA [38]. On the other hand, the pH measurements of scaffolds showed that the pH values decreased after immersion time, mainly due to a hydrolytic degradative process of PLA that included an attack on the ester bond via water molecules, breaking PLA into small parts and releasing the carboxylic group to the medium, reducing the pH values [54].

For the scaffolds with bioglass, the rate of degradation was higher compared to the same samples without n-BG but lower than for pure PLA. This indicates that the presence of PMMA has a reducing effect, but this is less with the presence of bioglass.

The presence of n-BG accelerated the degradation process; this phenomenon is due to the n-BG having improved the hydrophilicity of materials, increasing the water absorption capacity. However, the decrease in pH values due to the acidification of the medium was less pronounced. It could be explained as the release of alkaline ions from the nanoparticle surface, causing a buffered effect [29]. Recently, Cole et al., 2020, studied the mechanical and degradation properties of PMMA and borate BG (BBG) cements through different in vitro tests. They found that the addition of BBG in different concentrations is able to maintain the mechanical properties, and it controlled ion release for at least 21 days of study, which could improve the in vivo osteoconductive capacity of the material [55].

Finally, considering the results, it is important to highlight that the non-degradability of PMMA may be a limitation when considering bone regeneration therapies because it would imply a second operation to remove the implant. However, non-degradable polymers are widely used in other therapies as orthopedic substitutes for the treatment of bones, cartilage, and hips, as well as dental applications, standing out for their structural stability, as well as their mechanical resistance to wear [23,56]

### 2.8. In Vitro Cell Viability

The in vitro biological characterization of PP01, PP03, PPBG01, and PPBG03 scaffolds were analyzed regarding cell viability using the fetal human osteoblast-based cell line HBOF-1.19 after 1, 3, and 7 days of culture. The results presented in Figure 8A, corresponding to the statistical analysis, demonstrate that the scaffolds did not induce a reduction in more than 30% in viability on the study days compared to the cells alone used as a control, being viable substrates for their potential application [57]. It is possible to see that the behavior of the scaffolds had no significant variation concerning the control at all the times studied, showing adequate stability to promote cell adhesion. This was complemented with confocal microscopy images, where the cell nuclei can be observed to be in good shape, and they adhered to the surface of scaffolds (Figure 8B). Previous studies have indicated that both PLA and PMMA are matrices with good biological performance using different types of cell lines [16,54,58]. Xing et al., 2013, studied the response of osteoblasts in electrospun PMMA fibers bio-reinforced with HA. They found that these nanocomposites improved cell organization, increased the ALP activity, and accelerated osteoblast differentiation compared to pure PMMA fibers [59]. Rong et al. reported the good cell viability of electrospun PLA/PMMA matrices reinforced with hydroxyapatite nanoparticles using the MG-63 cell line [38]. The incorporation of n-BG also does not significantly modify the biological capacity of the scaffolds. Similar results were reported by Canales et al. when preparing PLA-based electrospun scaffolds with bioactive glass nanoparticles, finding that the presence of n-BG does not have a cytotoxic effect using cells derived from bone marrow of the ST-2 line. However, the nanoparticles did have a positive effect in promoting cellular differentiation towards bone lineage, demonstrated in a greater presence of ALP as a marker; this improvement is associated with the osteoinductive capacity of this type of nanoparticles [16]. Recently, Zaszczyńska et al., 2024, developed PMMA and n-HA electrospun fibers in different fiber orientations. They studied the in vitro biocompatibility of the scaffolds with human osteoblastic cells of the MG63 cell line. Their results indicated that all scaffolds, including those based on pure PMMA, do not exhibit cytotoxicity and increase cell viability. In addition, they concluded that the addition of nanoparticles improves cell–scaffold interaction, promoting better cell adhesion in the electrospun fibers [60]. These preliminary results suggest that PLA/PMMA/n-BG scaffolds are a viable alternative for application in bone tissue engineering.

### 2.9. In Vivo Biocompatibility

The in vivo biocompatibility of PP01, PP03, PPBG01, and PPBG03 scaffolds was evaluated via histological assays after 2 weeks of implantation. The macroscopic observations shown in Figure 9 revealed normal hair formation and utterly healthy skin for all samples. On the inner surface of the skin, the implantation sites of the nanofibers were identified, with no presence of exudates, signs of degradation, redness, or purulent discharge. All scaffolds implanted were encapsulated in the subdermal area as part of the response to tissue scarring due to the external material. The results are consistent with tissue healing, and the nanofiber fragment was observed, along with an inflammatory infiltrate. However, this situation should not be considered an indicator of a lack of biocompatibility but, rather, a slower reabsorption process [61]. On the other side, it is possible to observe that, for PLA after 2 weeks of implantation, a prominent inflammatory infiltrate was present, and the scaffolds are visible in Figure 9A,B. For PP03, the inflammatory response was slightly better than for PLA, showing signs of biodegradation (Figure 9C,D) and indicating that the incorporation of PMMA improves the healing process. Son et al. reported the preparation of PCL/PMMA electrospun scaffolds with different PCL/PMMA ratios for bone tissue engineering, and their results indicate that the relation of 7/3 has a good in vivo response, showing a null foreign response and better osteoinductive properties [25]. In addition, Cui et al., 2017, reported through in vitro and in vivo analyses the improved osteointegration ability of PMMA-based bone cements with the addition of Sr-BBG. For the in vivo assays, scaffolds were implanted into the medial tibial metaphysis of Sprague–Dawley rat tibiae for 8 and 12 weeks. The researchers found that Sr-BBG in PMMA stimulated new bone formation around the interface with the host bone at 8 and 12 weeks post-implantation, whereas PMMA only promoted the development of an intermediate layer of connective tissue [62]. On the other hand, histological analysis revealed an increased presence of blood vessels for scaffolds including n-BG (Figure 9E,F and Figure 9G,H, respectively). In both cases, the scaffolds exhibited an increase in the presence of blood vessels; in both cases, the scaffolds showed signals of neovascularization, and it was observed that, within the area of the fibrous scar, a healing process was taking place, characterized by the presence of repair cells such as fibroblasts and structures such as blood vessels necessary for the transport of cells and nutrients, as well as for the elimination of waste products, with the presence of a lymphocytic-type inflammatory infiltrate indicating a reduction in the response compared to samples without n-BG nanoparticles. Sharifi et al. reported the preparation of nanofibrous electrospun scaffolds based on gelatin and collagen reinforced with 4S5S bioglass nanoparticles and doped with Ag ions for tissue engineering. The in vivo analysis using dorsal subdermal implantation in BALC mice indicates that the presence of BG and Ag-BG reduces the inflammatory response, evidencing a lower presence of inflammatory cells on scaffolds with nanoparticles compared to the negative control, which was related to the positive effects of ion release of BG as Si, Ca, and Ag that improves the healing process [63]. The therapeutic effects of ions present on BG nanoparticles and the possibility to be doped with other kinds of positive ions allow the reduction in the typical inflammatory reaction related to the preceding body presence, being an excellent strategy to develop an exciting and novel biomaterial with an enhanced in vivo response with the presence of BG [64].

## 3. Materials and Methods

### 3.1. Materials

Poly (L-lactic acid) (PLLA) filament (Sakata 3D) with an average molecular weight of M_w_ = 335.173 g/mol and Poly (methyl methacrylate) (PMMA), from Sigma Aldrich, St. Louis, MO, USA, with an average molecular weight of M_w_ = 120,000 g/mol, were used. Bioactive glass nanoparticles (n-BG) were synthesized using the sol-gel method previously reported by Canales et al. [30].

### 3.2. Preparation of Hybrid PLA/PMMA and PLA/PMMA/n-BG Fibers via Electrospinning Technique

PLA/PMMA hybrid fibers were prepared in different ratios, 100/0, 75/25, 50/50, 25/75, and 0/100 in %wt./wt. The solutions were prepared at 12.5% wt./v, using a mixture of chloroform (CF) (Winkler; purity: 99.8%) and dimethylformamide (DMF) (Merck, Boston, MA, USA; purity: 99.9%) as a solvent in a 9:1 volume ratio, respectively.

In the first stage, the amount of PLA and PMMA was weighed and dissolved in the mixture (CF/DMF) and stirred for 24 h. For the PLA/PMMA/n-BG solutions, a 10% load of nanoparticles was incorporated concerning the PLA/PMMA mixture. Before the preparation of the solutions, the corresponding amount of nanoparticles was added in 5 mL of CF and sonicated for 1 h to reduce the agglomeration phenomenon. Finally, the appropriate amount of the PLA/PMMA mixture was added, the volume was completed, and electrospinning was done using the same parameters for the PLA/PMMA mixture without nanoparticles. The prepared samples were specified using simpler codes to facilitate their understanding; these codes and the correct quantities and ratios are detailed in Table 2.

The electrospinning process was carried out using TongLi-Tech (TL-01) equipment, with a voltage of 20 KV and a flow of 1 mL/hour at room temperature. The solutions were deposited in a 20 mL syringe using a size-22 needle and a collector needle distance of 15 cm. The developed electrospun fibers were collected on metallic paper, labeled, and stored under refrigeration.

### 3.3. Characterization of n-BG Nanoparticles, Hybrid PLA/PMMA, and PLA/PMMA/n-BG Fibers

#### 3.3.1. Nanoparticle Characterization

The morphology and size of the n-BG were analyzed via Transmission Electron Microscopy (TEM) (Hitachi, model HT7700, Tokyo, Japan) using the ImageJ software 1.53. The diameter was determined by considering a total of 60 measurements. In addition, they were chemically characterized via FT-IR (Perkin Elmer, Spectrum Two, Waltham, MA, USA) and structurally via X-ray Diffraction (XDR) (Bruker, D8 advance, Billerica, MA USA).

#### 3.3.2. Morphology and Size of Hybrid Electrospun Fibers

The morphology of hybrid PLA/PMMA and PLA/PMMA/n-BG fibers was analyzed using Scanning Electron Microscopy (FE-SEM) (Zeiss, Gemini SEM360, Oberkochen, Germany). Initially, the samples were coated with gold to increase their conductivity (~5 nm); the average was measured from the SEM images obtained using the ImageJ software 1.53 (Java, Bethesda, MD, USA). This average was calculated by considering 200 measurements at different randomly selected points of the fibers.

#### 3.3.3. Wettability of the Hybrid Electrospun Fibers

The contact angle (CA) method was used to determine the surface wettability properties of the developed fibers. A tensiometer (Biolin Scientific, model ThetaLite, Gothenburg, Sweden) measured the contact angle with a 4 µL drop of distilled water in an air medium. The camera took images at 10 Hz for 10 s, resulting in 100 images on average. The measurement was repeated five times, and the contact angle was calculated by fitting a Young–Laplace surface to the droplet contour.

#### 3.3.4. Infrared Spectroscopy Analysis of the Hybrid Electrospun Fibers

The chemical composition of the electrospun fibers was analyzed via Fourier Transform Infrared Spectroscopy (FT-IR) (Perkin Elmer, Spectrum Two, Waltham, MA, USA) to obtain a typical peak associated with PLA, PMMA, and n-BG.

#### 3.3.5. Mechanical Characterization of Hybrid Fibers

The mechanical behavior of the developed hybrid fibers was analyzed through Young’s modulus, tensile strength, and elongation at break via a uniaxial tension test with biaxial equipment (Cell Scalle, Biotester 300, Waterloo, ON, Canada). Five strips for each fiber of 10 × 5 mm were cut in different orientations of mats, based on a standard tensile testing ASTM D882-18, for the plastic film below 1 mm of thickness [65]. The test was evaluated using a cell load of 2.5 N to a rate of 0.5 mm/min.

#### 3.3.6. In Vitro Bioactivity in Simulated Body Fluid (SBF)

The bioactivity ability of the hybrid fibers was analyzed using Simulated Body Fluid (SBF) after 7, 14, 21, and 28 days of immersion. For this analysis, a rectangular of 4 × 1 cm^2^ was cut and immersed into 40 mL of SBF in a falcon tube of 50 mL, according to the relation biomaterial/volume 1 cm^2^/10 mL reported by Kokubo et al. [66]. The tube was put under stirring at 37 °C in an incubator during the time of assays (Benchmark, Incu-shaker 10 L, Sayreville, NJ, USA). After the time of immersion, the samples were removed from the falcon tube, washed with water, and wiped carefully. The fibers were dried at 25 °C for 12 h. Finally, the fibers were characterized using FT-IR (Perkin Elmer, Spectrum Two, Waltham, MA, USA), and FE-SEM (Zeiss, Gemini SEM360, Oberkochen, Germany) was used to study the formation of an apatite layer on the surface of the fibers.

#### 3.3.7. In Vitro Hydrolytic Degradation in Phosphate-Buffered Saline Solution (PBS)

The hydrolytic degradation rate of the hybrid PLA/PMMA and PLA/PMMA/n-BG fibers was analyzed using a phosphate-buffered saline solution (PBS) after 7, 14, 21, and 35 days of immersion. A rectangle of 4 × 1 cm^2^ was cut and immersed into 40 mL of PBS, following the same biomaterial/volume relation of the bioactivity analysis [67]. After immersion, the samples were removed from the falcon tube, washed with water, and dried at 25 °C for 12 h in a vacuum camera. Finally, the degradation process was studied using FT-IR (Perkin Elmer, Spectrum Two, Waltham, MA, USA), FE-SEM (Zeiss, Gemini SEM360, Oberkochen, Germany), and a pH-meter (Hanna Instruments, HI-3222, Woonsocket, RI, USA).

#### 3.3.8. In Vitro Cell Viability

The cell viability of the developed scaffolds was analyzed using the human fetal osteoblast cell line HBOF-1.19. First, cells were cultured in a DPMI medium supplemented with 10% *v*/*v* fetal bovine serum and 1% *v*/*v* penicillin–streptomycin in cell culture tubes and incubated at 37 °C in a controlled atmosphere of 5% CO_2_ and 95% relative humidity for 24 h; the cells were used with 60–70% of confluency. The samples were previously fixed on specialized cell-holder supports obtained via 3D printing using PLLA as filament. These cell holders were previously sterilized for 24 h in 70% ethanol and with ultraviolet radiation for 1 h on both sides. Then, 15 × 15 mm^2^ from each scaffold was cut, assembled, and fixed in the holder and placed in the 24-well plate for subsequent cell seeding. A density of 45,000 cell/cell-holder was seeded and incubated at 37 °C for 1, 3, and 7 days, using as a control single cells in the presence of a medium via a colorimetric assay using the MTT reagent 3-(4,5-dimethylthiazol-2-yl)-2,5-diphenyltetrazolium bromide (Merck, Boston, MA, USA). The reagent was dissolved in a supplemented MEM medium in a 1:10 solution; 400 µL of said solution was added to each well and incubated for 3 h in the dark at 37 °C. After the incubation time had elapsed, the supernatant was removed, and 400 µL of DMSO was added to be shaken for 6 min and later transferred to a 96-well plate for measurement in an Infinite 150 ELISA reader (Tecan, Männedorf, Switzerland) at 490 nm.

#### 3.3.9. In Vivo Biocompatibility

To evaluate the in vivo biocompatibility of the scaffolds, a subdermal dorsal implantation model was studied in adult mice according to ISO 10993-6 [68]. Fifteen male Balb/c mice aged 8–12 weeks were employed as experimental subjects. The animals were sedated via the intraperitoneal administration of a solution of ketamine at 30 mg/kg and xylazine at 70 mg/kg. Each animal underwent a surgical implantation of electrospun membranes in the dorsal subdermal area. In the surgical preparations, a nanocomposite film with dimensions of 10 mm in length by 5 mm in width, corresponding to each of the four nanocomposite samples and the control without (S1–2 and S4–5), was inserted.

#### 3.3.10. Statistical Analysis

Statistical data analysis was conducted using the one-way analysis of variance (ANOVA) with the program *GraphPad Prism 5.0.* To evaluate the statistically significant difference between groups, the Bonferroni test was applied to counter the multiple comparisons of the null hypotheses. Values of * = *p* > 0.05, ** = *p* > 0.01, and *** = *p* > 0.001 were considered significant.

## 4. Conclusions

The preparation of hybrid PLA/PMMA/n-BG electrospun scaffolds using chloroform/dimethylformamide in a relation of 9:1 as the solvent system was successful in developing a porous and interconnective matrix for potential application in bone tissue engineering. The incorporation of PMMA had a significant effect on the morphology and fiber diameter compared to pure PLA, increasing the presence of beads with the increase in PMMA content, with the PP02 as the optimal relation. In addition, the incorporation of n-BG also increased the beads, which was related mainly to nanoparticle agglomeration. The hydrophobic character of the material was not modified with the presence of PMMA or n-BG; however, the mechanical behavior changes became more flexible, showing a decrease in Young’s modulus, as well as the tensile strength and elongation at break. The presence of bioglass nanoparticles achieves the bioactive properties of PLA-based scaffolds, showing the ability to promote the growth of the hydroxyapatite layer on the fiber surface, as was corroborated using SEM images. The hydrolytic degradation of scaffolds was retarded compared to neat PLA, which was mainly related to the incorporation of PMMA. Finally, an in vitro biological evaluation with the HBOF-1.19 cell line indicated that PLA/PMMA/n-BG did not exhibit cytotoxic effects and promoted cell adhesion. Furthermore, the scaffolds did not affect the healing process, which was confirmed with an in vivo analysis using a dorsal subdermal model, demonstrating the materials’ biocompatibility and making them a promising material for bone tissue engineering.

## Figures and Tables

**Figure 1 ijms-25-06843-f001:**
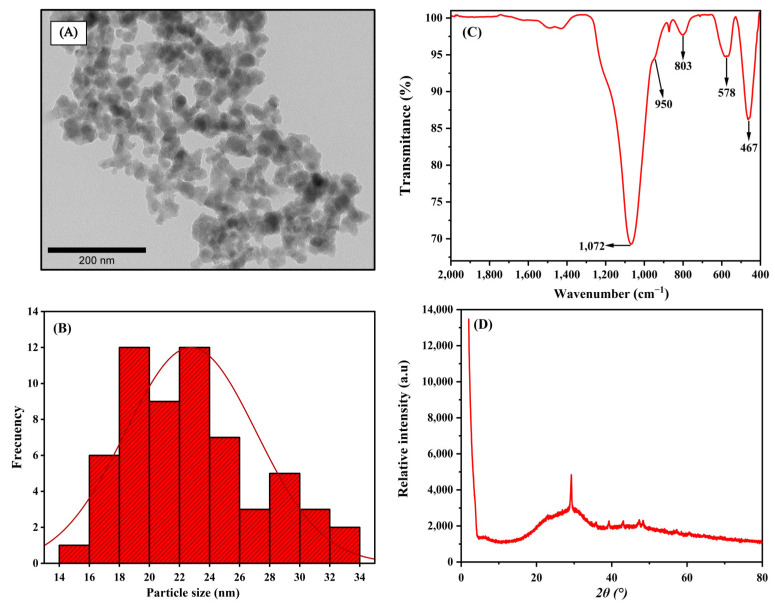
Bioglass nanoparticle characterization. (**A**) TEM image, (**B**) histogram of size distribution, (**C**) FT-IR analysis, angd (**D**) XRD analysis.

**Figure 2 ijms-25-06843-f002:**
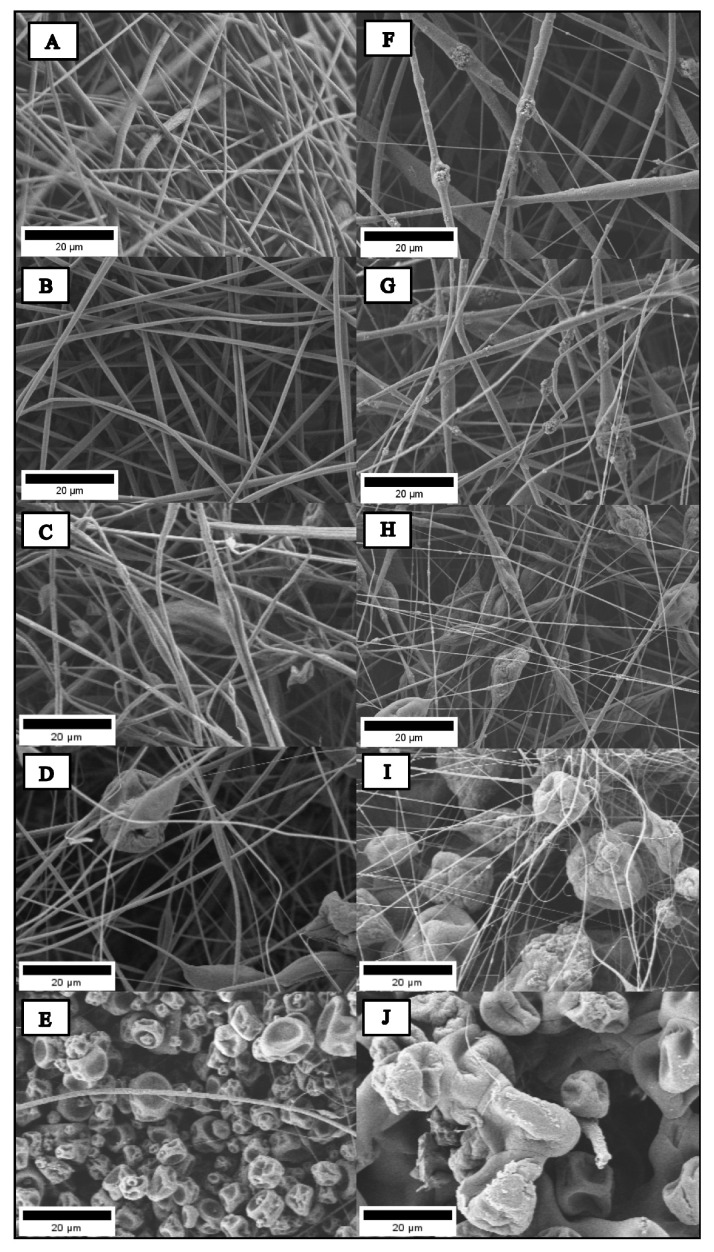
SEM images of hybrid electrospun fibers (**A**) PP01, (**B**) PP02, (**C**) PP03, (**D**) PP04, (**E**) PP05, (**F**) PPBG01, (**G**) PPBG02, (**H**) PPBG03, (**I**) PPBG04, and (**J**) PPBG05.

**Figure 3 ijms-25-06843-f003:**
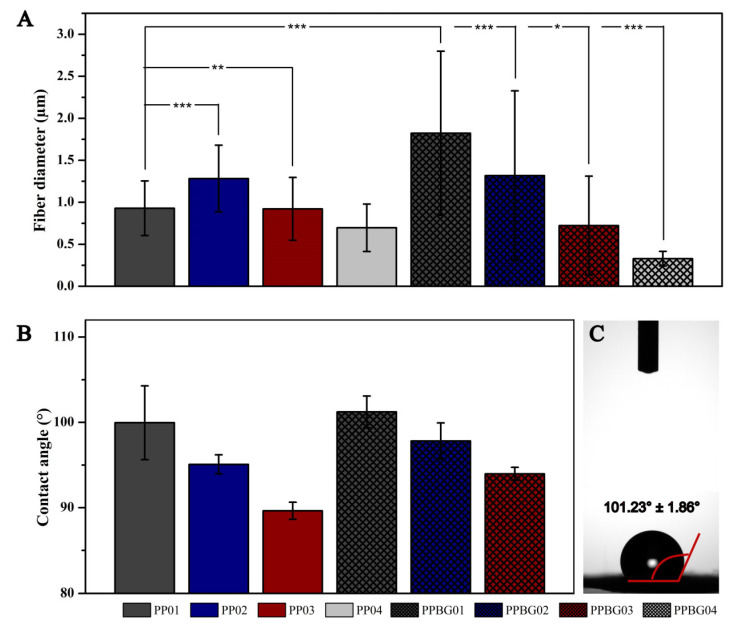
Statistical analysis of fiber diameter and contact angle of scaffolds without and with 10 wt.% of n-BG. (**A**) fiber diameter; (**B**) contact angle and (**C**) sample CA PPBG01. (n_diameter_ = 50; n_contac angle_ = 20, *** = *p* < 0.001, ** *p* < 0.01, and * = *p* < 0.05).

**Figure 4 ijms-25-06843-f004:**
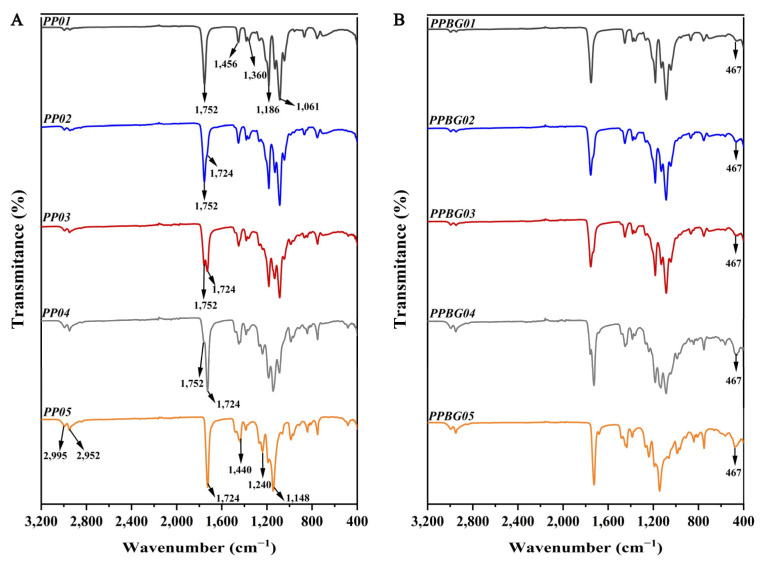
Infrared spectroscopy analysis of electrospun fibers (**A**) without and (**B**) with bioglass nanoparticles.

**Figure 5 ijms-25-06843-f005:**
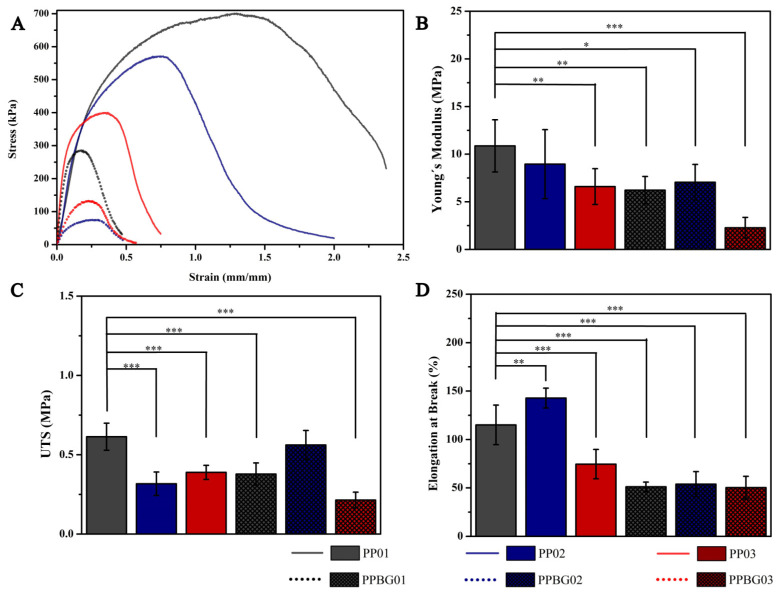
(**A**) Stress–strain diagram and statistical analysis of mechanical properties of scaffolds, (**B**) Young’s modulus, (**C**) tensile strength, and (**D**) elongation at break (n = 7, *** = *p* < 0.001, ** *p* < 0.01, and * = *p* < 0.05).

**Figure 6 ijms-25-06843-f006:**
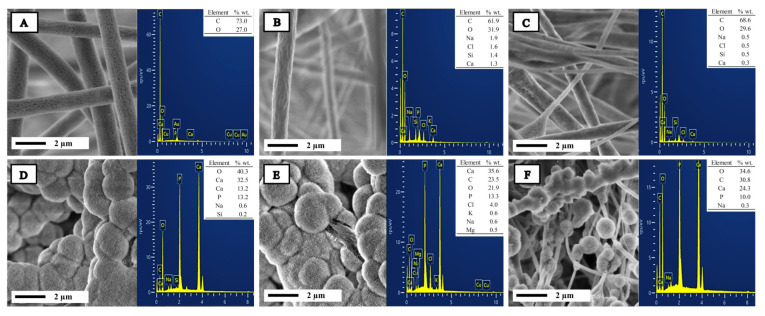
SEM images and EDS analysis of scaffolds in SBF solution after 21 days of immersion for (**A**) PP01, (**B**) PP02, (**C**) PP03, (**D**) PPBG01, (**E**) PPBG02, and (**F**) PPBG03.

**Figure 7 ijms-25-06843-f007:**
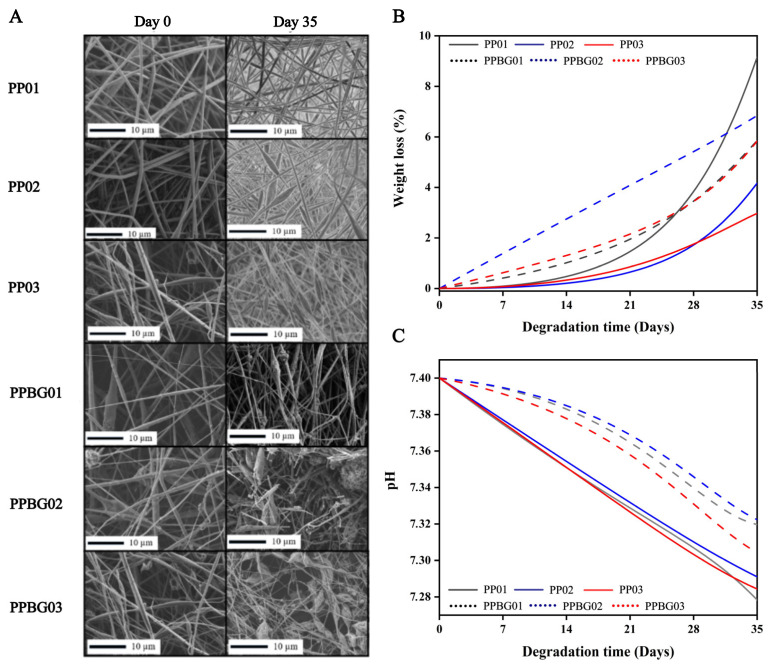
Hydrolytic degradation analysis after 35 days of immersion in PBS solution at 37 °C. (**A**) SEM images, (**B**) weight loss, and (**C**) pH measurements vs. degradation time curves of scaffolds.

**Figure 8 ijms-25-06843-f008:**
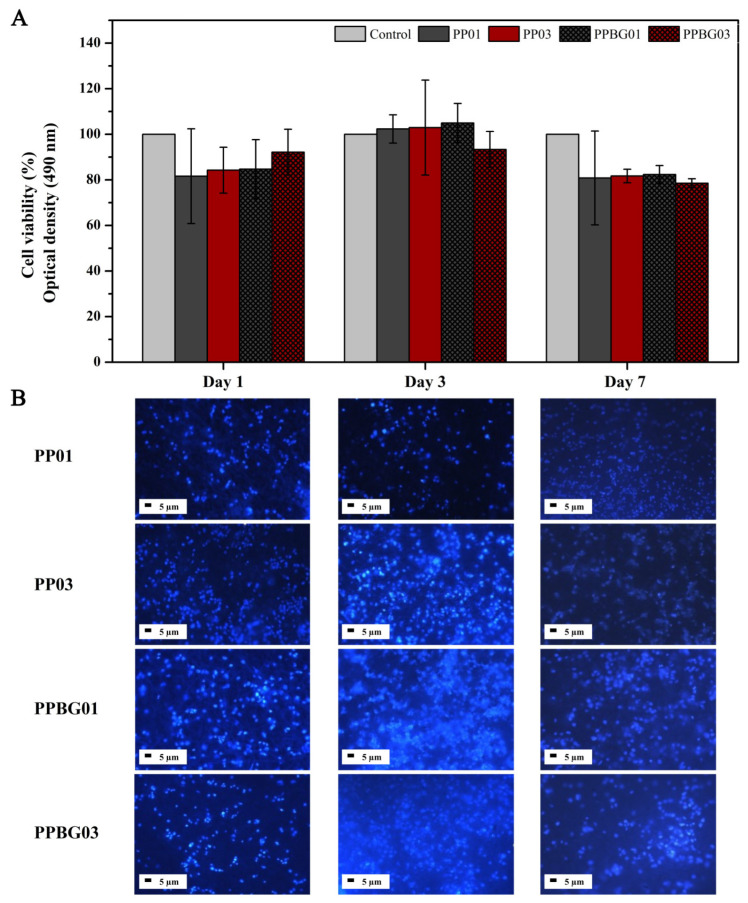
In vitro cell viability of scaffolds after 1, 3, and 7 days of cell culturing via a colorimetric assay using the MTT reagent in a DMEM medium (n = 3). (**A**) Statistical analysis of cell viability using ANOVA analysis and Bonferroni posttreatment for significance and (**B**) Fluorescence Microscopy using a DAPI as a cell nucleus marker.

**Figure 9 ijms-25-06843-f009:**
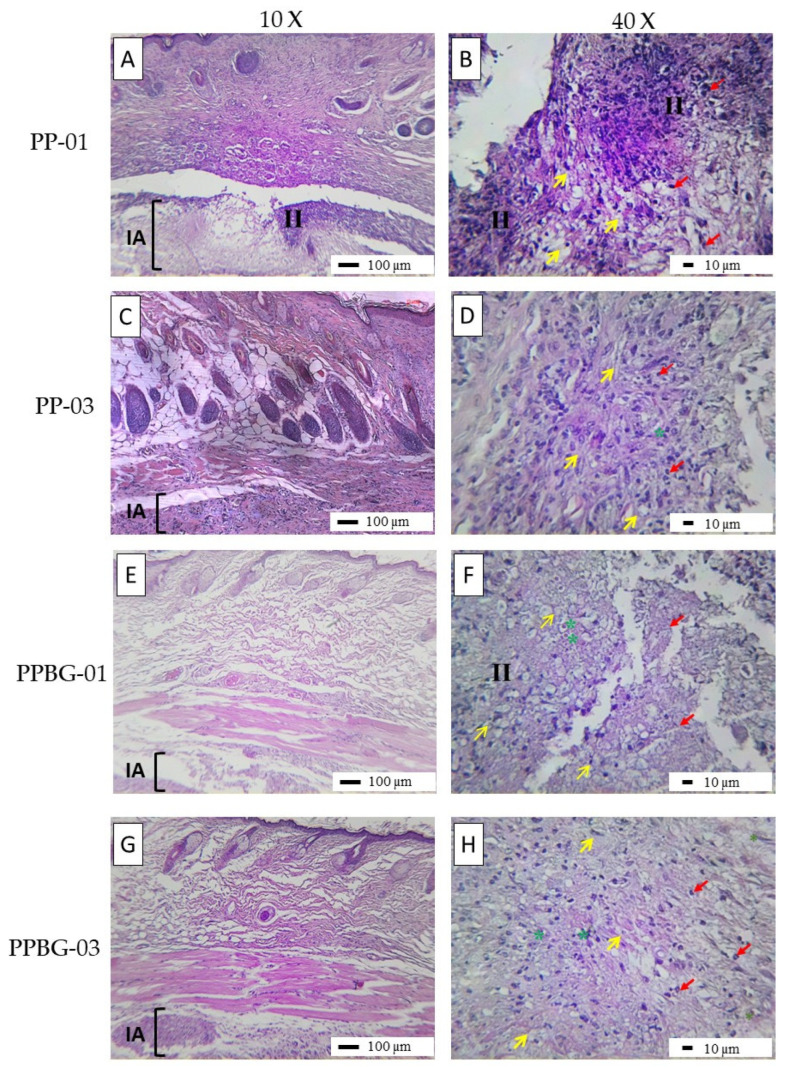
Histological image of in vivo biocompatibility assay for (**A**,**B**) PP-01, (**C**,**D**) PP-03, (**E**,**F**) PPBG-01, and (**G**,**H**) PPBG-03 via dorsal subdermal implants in BALB mice at 2 weeks with a magnification of 10 and 4×. IA = implanted area; II = Inflammatory infiltatre; yellow arrow = scaffold material; green asterisk = blood vessels; red arrow = leukocytes.

**Table 1 ijms-25-06843-t001:** Ca/P relation of scaffolds without and with 10 wt.% of n-BG obtained via SEM-EDS analysis of the hydroxyapatite (HA) growth zones after 21 days of immersion of SBF solution.

Sample	Ca (% wt./wt.)	P (% wt./wt.)	Ca/P Relation
PP01	0	0	0
PP02	0	0	0
PP03	0	0	0
PPBG01	32.5	13.3	1.90
PPBG02	35.5	13.2	2.06
PPBG03	24.3	10.0	1.87

**Table 2 ijms-25-06843-t002:** Amounts, ratios, and codes used in the preparation of PLA/PMMA electrospun fibers.

Sample	Code	Amount PLA [g]	Amount PMMA [g]	Amount n-BG [g]
Neat PLA	PP-01	1.250	0	0
PLA/PMMA (75/25)	PP-02	0.9375	0.3125	0
PLA/PMMA (50/50)	PP-03	0.6250	0.6250	0
PLA/PMMA (25/75)	PP-04	0.3125	0.9375	0
Neat PMMA	PP-05	0	1.250	0
PLA + 10 wt.% n-BG	PPBG-01	1.125	0	0.1250
PLA/PMMA (75/25) + 10 wt.% n-BG	PPBG-02	0.8437	0.2813	0.1250
PLA/PMMA (50/50) + 10 wt.% n-BG	PPBG-03	0.5625	0.5625	0.1250
PLA/PMMA (25/75) + 10 wt.% n-BG	PPBG-04	0.2813	0.8437	0.1250
PMMA + 10 wt.% n-BG	PPBG-05	0	1.125	0.1250

## Data Availability

Not available.

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
