# Peer review of "Development of Bioactive Hybrid Poly(lactic acid)/Poly(methyl methacrylate) (PLA/PMMA) Electrospun Fibers Functionalized with Bioglass Nanoparticles for Bone Tissue Engineering Applications"

_ijms, 2024, doi:10.3390/ijms25136843_

Round 1

Reviewer 1 Report

Comments and Suggestions for Authors

The authors performed detailed characterization of scaffolds based on PLA, PMMA and n-BG which can help i further development of these materials. However, they have to revise their manuscript thoroughly to be more understandable and attractive to readers.  I left my comments and remarks in pdf file attached below. 

Comments on the Quality of English Language

Author Response

Reviewer 1.

  1. Maybe 'ratio' is more convenient word then 'relation'.

Response: Thanks for your suggestion.

 In text line 24, we change the word relation by ratio.

  1. This sentence has to be written more clearly.

PLA and PLA/PMMA scaffolds were developed obtaining a randomly homogeneous fiber deposition generating a porous structure with a pore interconnectivity, neat PMMA scaffold did not develop with the solvent system used.

Response: Thanks for your suggestion, to be more precise with the idea that we want to express, a new sentence was incorporated in the lines 27-30.

Neat PLA and PLA/PMMA hybrid scaffolds were developed successfully by (CF:DMF) solvent system, obtaining a randomly fiber deposition that generated a porous structure with pore interconnectivity. However, with the solvent system used, it was not possible to generate fibers in the case of the neat PMMA sample.

  1. Increases the presence of beads WHICH IS associated with low interaction... If this is what the author wanted to say, or the sentence should be revised otherwise.

Response: Thanks, effectively that is the main idea that we try to say, however, according with your suggestion, we incorporated a new sentence in the lines 30-33.

With the increase of amount of PMMA in PLA/PMMA ratios, the fiber diameter of hybrid scaffolds decreases and increases the defects (beads) in the fiber structure, these beads are associated to a nanoparticle agglomeration, that could be related with a low interaction between n-BG and the polymer matrix

  1. n-BG the abbreviation should be defined.

Response: Thank you, the n-BG abbreviation was defined in the previous sentences in the line 25.

Hybrid scaffolds based on PLA and PLA/PMMA with 75/25, 50/50 and 25/75 weight ratio and functionalized with 10 wt.% of bioglass nanoparticles (n-BG).

  1. The PLA/PMMA/n-BG showed a good in vitro and in vivo biological behavior, not showing a cytotoxic effect being a suitable substrate to cell adhesion using a HBOF-1.19 cell line and dorsal subdermal model in BLC male mice. This sentence should be revised to clearly indicate the results of in vitro and in vivo investigations, this way it looks like only cytotoxic effect was investigated.

Response: Thank you, by the in vitro and in vivo analysis we try to evaluate the cytotoxic effect of the materials against a cell line HBOF-1.19 and also in physiological environment by subdermal model, as a preliminary analysis, that could be bring us an initial idea on their biological behavior and their future possible application on clinical procedures for tissue engineering.

However, taking your suggestion to enhance the redaction, the next sentence was incorporated in the text between the lines 39-44.

The in vitro analysis using a HBOF-1.19 cell line indicate that the PLA/PMMA/n-BG scaffold showed a good cell viability and is able to promote the cell proliferation after 7 days. For other side, the in vivo biocompatibility evaluated by subdermal model in BALC male mice, corroborate the good behavior of the scaffolds not generating a cytotoxic effect and being able to enhance the healing process, suggesting that the materials are suitable for their potential application in tissue engineering.

  1. to provide materials that can evaluate, cure, replace, or regenerate any tissue, organ, or human body function,

Response: Thanks, the word evaluate was erase of the text.

  1. Its interest in biomedical and tissue engineering areas is given by its excellent biocompatibility. The sentence should be revised to be grammatically correct.

Response: Thank you, the sentences remarked was changed for the next sentence in the lines 63-64.

Its interest in biomedicine and tissue engineering is due to its excellent biocompatibility and biodegradability.

  1. It is non-toxic, allowing it to have potential use in the human body.

Response: Thanks for your recommendation, we taken the sentence proposed by you and was incorporated in the text in the lines 65-66.

It is non-toxic, which enables its potential application in the human body.

  1. in some areas; this is related to its high rigidity. The sentence should be revised to be grammatically correct.

Response: Thanks for your recommendation, we taken the sentence proposed by you and was incorporated in the text in the line 67.

In some areas related to its high rigidity.

  1. Maybe 'absence' is more convenient word then 'not having'.

Response: Thanks for your suggestion.

In text line 68, we change the word not having by absence.

  1. Grammatical errors in words "shows", "promotes", "show", "of", "on", "presenting" and "present".

Response: Thanks for your suggestion.

Any grammatical errors found in the text have been corrected.

  1. Maybe ‘presenting’ is more convenient word then 'with'.

Response: Thanks for your suggestion.

In text line 160, we change the word presenting by with.

  1. The wettability of a material, determined by contact angle. Use this abbreviation CA later, so it has to be defined.

Response: Thanks for your recommendation, we taken the sentence proposed by you and was incorporated in the text in the lines 215.

The wettability of a material, determined by contact angle (CA).

  1. Previous studies show the hydrophilic nature of PMMA, presenting a contact angle value close to 67.8 ± 1.4. This nature is the main reason for the decrease in CA value that increases the PMMA content in the scaffolds. The sentence has to be revised.

Response: Thank you, we took into account your suggestion to improve the writing, the following sentence was incorporated into the text between lines 231 - 234.

This phenomenon is mainly due to the hydrophilic nature of PMMA, resulting in a decrease in the CA value as its content in the scaffolds increases. Previous studies show the hydrophilic nature of PMMA, presenting a contact angle value close to 67.8 ± 1.4.

  1. The diameter and imperfection of the fibers the two factors that are the main reason for the decrease in tensile strength.

Response: Thank you, the mechanical properties of the fibers depend on many factors, among them the increase in diameter and the appearance of imperfections in the fibers decrease the elastic modulus, tensile strength and elongation at break. To better understand the presentation and analysis of the results of the mechanical properties, new references have been included, where they investigate the effect of various factors on the mechanical properties of the fibers. Likewise, two specific studies were added on the effect of fiber diameter and imperfections on mechanical properties. These changes have been incorporated between lines 310 – 326.

Liverani et al. reported similar results when incorporating 30% n-BG to mixtures of PCL and chitosan. They found that adding n-BG significantly decreased the mechanical proper-ties, due to the increase in the diameter of the fibers and the appearance of defects in the scaffolds [48]. Currently, the effect of many parameters on the mechanical properties of electrospun scaffolds has been studied [49-51]. Alharbi et al, studied the effect of molecular weight and fiber diameter on the mechanical properties of PCL scaffolds. They obtained that the elastic modulus decreased drastically with the increase in fiber diameter, while there were no significant variations with the change in molecular weights of PCL [52]. While Rashid et al, concluded that the appearance of defects such as beads in fibers alters their mechanical properties by creating stress concentration points and structural discontinuities, leading to premature fracture and unequal load distribution [53]. These defects also generate weak interfaces with the polymer matrix, reducing tensile strength and elasticity, and in general, decreasing the load capacity and deformability of the fibers [49,50]. Similarly in our study, scaffolds that included PMMA and n-BG showed larger fiber diameters and the appearance of imperfections, these two factors being the main reasons related to the decrease in mechanical properties.

  1. Canales et al. reported the preparation of electrospun scaffolds based on Poly (lactic acid) PLA nanocomposites,

Response: Thanks, the word Poly (lactic acid) is not necessary, so it has been removed from the text.

  1. the Ca/P relation obtained for the growth hydroxyapatite zones mapped.

Response: Thanks, the sentence remarked was changed for the text between lines 352 - 353.

 the Ca/P ratio obtained for the samples studied by EDS analysis.

  1. Scanning Electron Microscope (SEM) the abbreviation was introduced previously.

Response: Thanks, the abbreviation SEM was defined previously, so it has been removed from the text.

  1. 2.7.A shows the SEM images of each sample for 0 and 35 days, respectively, it can be observed that PLA and PLA/PMMA without the presence of bioglass, the degradation process is not detectable on the surface of the fibers, from the mass loss curve Fig. 2.7.B the scaffold based on pure PLA reached 9%, and for the PLA/PMMA mixtures (75/25) and (50/50), the values ​​were 4 and 3 % , respectively, These indicate that the incorporation of PMMA reduces the degradation rate of PLA. The sentence is too long and barely understandable.

Response: Thank you very much for the observation, the sentence has been modified to understand it more clearly between lines 369-374.

Figure 2.7.A shows the SEM images of the samples after 0 (left) and 35 (right) days of study. These results revealed that the samples with n-BG degraded faster compared to the samples without nanoparticles. In terms of mass loss (Figure 2.7.B), it was observed that the PP01 scaffold experienced a 9% loss, while the PP02 and PP03 samples lost 4% and 3%, respectively. These results indicate that the inclusion of PMMA reduces the degradation rate of pure PLA.

  1. but less than pure PLA.

Response: Thanks for your recommendation, we have replaced the phrase "less than" with "is lower than" in text line 383.

  1. but this is less with the presence of bioglass.

Response: Thanks for your observation, we have replaced the phrase "this is less with the" with "it is less pronounced in the" in text line 384.

  1. than PLA showing signs of biodegradation Fig. 2.9.C-D indicating that.

Response: Thanks for your recommendation, we have corrected the sentence between lines 452-453 of the text.

than for PLA showing signs of biodegradation (Fig. 2.9.C-D) and indicating that.

  1. In addition, for PLA and PLA/PMMA with n-BG 10 wt.% Fig. 2.9.E-F and G-H, respectively, the histological analysis showed an increase in the presence of blood vessels, in both cases, the scaffolds showed signals of neovascularization and observed that within the area of the fibrous scar, a healing process was taking place, characterized by the presence of repair cells such as fibroblasts, structures such as blood vessels necessary for the transport of cells and nutrients, as well as for the elimination of waste products, with the presence of a lymphocytic-type inflammatory infiltrate indicating a reduction in the response compared to samples without Bioglass nanoparticles. Too long sentence, very hard to follow.

Response: Thank you very much for the observation, the sentence has been modified to understand it more clearly between lines 463 - 471.

On the other hand, histological analysis revealed an increased presence of blood vessels for scaffolds including n-BG (Fig. 2.9. E-F and G-H, respectively). In both cases, the scaffolds exhibited  an increase in the presence of blood vessels, in both cases, the scaffolds showed signals of neovascularization and observed that within the area of the fibrous scar, a healing process was taking place, characterized by the presence of repair cells such as fibroblasts, structures such as blood vessels necessary for the transport of cells and nutrients, as well as for the elimination of waste products, with the presence of a lymphocytic-type inflammatory infiltrate indicating a reduction in the response compared to samples without n-BG nanoparticles.

  1. To determine the in vivo biocompatibility of developed scaffolds, a dorsal subdermal implantation method was studied under ISO 10993-6 standard guides the performance of dorsal subcutaneous implantations in adult mice to study biocompatibility in vivo. The sentence should be revised, some phrases are repetitive, and it is not grammatically correct.

Response: Thank you very much for the recommendation, the phrase has been reviewed and modified for your better understanding. The sentence is found between lines 596 - 597 of the text.

To evaluate the in vivo biocompatibility of the scaffolds, a subdermal dorsal implantation model was studied in adult mice according to ISO 10993-6.

  1. Maybe ‘concerning’ is more convenient word then 'compared to'.

Response: Thank you, in text line 615, we change the word concerning by compared to.

  1. Finally, the in vitro biological characterization using a HBOF-1.19 cell line indicates that the PLA/PMMA/n-BG did not show a cytotoxic effect and promote cell adhesion, and the in vivo analysis using a dorsal subdermal model corroborates that scaffolds have biocompatibility and did not affect the healing process being a suitable material for bone tissue engineering. It is not clear to me from this sentence if PLA-PMMA/n-BG did not show a cytotoxic effect, but it promoted cell adhesion, or it did not show the cytotoxic effect neither promoted cell adhesion. The sentence it is not clear, PLA-PMMA/n-BG did not show cytotoxic effect but promoted cell adhesion, or did not show cytotoxic effect and did not promote cell adhesion?

Response: Thank you very much for the observation, in fact the phrase was confusing regarding the biological results. What we want to indicate is that the scaffolds did not show cytotoxic effects, while they promoted cell adhesion during the in vitro study. That is why we have corrected the text to present our conclusions more clearly. The corrected sentences are found in lines 624 - 628 of the text.

Finally, in vitro biological evaluation with the HBOF-1.19 cell line indicated that PLA/PMMA/n-BG did not exhibit cytotoxic effects and promoted cell adhesion. Furthermore, the scaffolds did not affect the healing process, which was confirmed by in vivo analysis using a dorsal subdermal model, demonstrating their biocompatibility and making them a promising material for bone tissue engineering.

  1. The Figure 2.9 is too small, details cannot be seen.

Response: Thanks for your observation, the figure was not clear enough. We have made changes to improve clarity and level of detail.  The new figure was incorporated in the text using magnification of 10 and 40 X.

Reviewer 2 Report

Comments and Suggestions for Authors

The manuscript entitled “Development of bioactive hybrid poly (lactic acid)/poly (methyl 2 methacrylate) (PLA/PMMA) electrospun fibers functionalized with bioglass nanoparticles for bone tissue engineering applications” demonstrates the development of hybrid scaffolds using PLA and various combinations of PLA/PMMA alongside incorporation of bioglass nanoparticles. The developed scaffolds have been indicated as promising scaffolds promoting in vitro cell adhesion without cytotoxicity and exhibited biocompatibility in vivo. The overall experimental design and research value exhibits potential in promoting bone tissue engineering. However, after through review of the article, I have found that manuscript needs vigorous modifications and many changes before the consideration for publication. Please revise the manuscript according to the changes suggested below;

1.             The manuscript has multiple redundancies throughout. The authors are advised to revise the whole manuscript, please.

2.            It seems like that author have discussed only 2-4 references throughout the manuscript. Such as Canales et al, Son et al and Rong et al etc. Please enrich the discussion of every figure with most relevant and more numbers of recent studies.

3.            The short names of several groups such as PLA/PMMA (50/50) or PLA/N-BG 10 wt% etc. should be changed to short abbreviations, similarly, the very long sentences in the introduction and results section should be revised for more clarity and explanation.

4.             In Figure 2.8, the graph for cell viability can be changed to single graph for day 1, 3 and 7, rather than three separate graphs. Secondly, please mention the procedure, how cell viability (%) was calculated. In figure 2.8 legend DNEM should be replaced to DMEM

5.            Please mention the limitations of the current study and its future implications in the discussion section.

6.            It is suggested to completely write the text for every figure first, then add the figure at last rather than describing the results after figure legend etc.  

Comments on the Quality of English Language

 The manuscript has multiple redundancies throughout. The authors are advised to revise the whole manuscript, please.

Author Response

Reviewer 2

Comments and Suggestions for Authors

The manuscript entitled “Development of bioactive hybrid poly (lactic acid)/poly (methylmethacrylate) (PLA/PMMA) electrospun fibers functionalized with bioglass nanoparticles for bone tissue engineering applications” demonstrates the development of hybrid scaffolds using PLA and various combinations of PLA/PMMA alongside incorporation of bioglass nanoparticles. The developed scaffolds have been indicated as promising scaffolds promoting in vitro cell adhesion without cytotoxicity and exhibited biocompatibility in vivo. The overall experimental design and research value exhibits potential in promoting bone tissue engineering. However, after through review of the article, I have found that manuscript needs vigorous modifications and many changes before the consideration for publication. Please revise the manuscript according to the changes suggested below.

  1. The manuscript has multiple redundancies throughout. The authors are advised to revise the whole manuscript, please.

Response: Thanks for your suggestion, to improve the manuscript we revised and modified the article, all these modifications were marked in blue in the text.

  1. It seems like that author have discussed only 2-4 references throughout the manuscript. Such as Canales et al, Son et al and Rong et al etc. Please enrich the discussion of every figure with most relevant and more numbers of recent studies.

Response: Thanks for your suggestion, effectively we try to compare our results with works that using a similar polymers system, but not exist a lot of reports about PLA/PMMA. However, we take you recommendation and others work related to electrospun fibers based on PLA, PMMA or another polymer matrix and was incorporated to improve the article.

These incorporations are:

Mechanical section:

Between lines 310 to 326: Liverani et al. reported similar results when incorporating 30% n-BG to mixtures of PCL and chitosan. They found that adding n-BG significantly decreased the mechanical properties, due to the increase in the diameter of the fibers and the appearance of defects in the scaffolds [49]. Currently, the effect of many parameters on the mechanical properties of electrospun scaffolds has been studied [50-52]. Alharbi et al, studied the effect of molecular weight and fiber diameter on the mechanical properties of PCL scaffolds. They obtained that the elastic modulus decreased drastically with the increase in fiber diameter, while there were no significant variations with the change in molecular weights of PCL [53]. While Rashid et al, concluded that the appearance of defects such as beads in fibers alters their mechanical properties by creating stress concentration points and structural discontinuities, leading to premature fracture and unequal load distribution [54]. These defects also generate weak interfaces with the polymer matrix, reducing tensile strength and elasticity, and in general, decreasing the load capacity and deformability of the fibers [55,56]. Similarly in our study, scaffolds that included PMMA and n-BG showed larger fiber diameters and the appearance of imperfections, these two factors being the main reasons related to the decrease in mechanical properties.

  1. Salimbeigi, G., Cahill, P. A., & McGuinness, G. B. (2022). Solvent system effects on the physical and mechanical properties of electrospun Poly (ε-caprolactone) scaffolds for in vitro lung models. Journal of the Mechanical Behavior of Biomedical Materials, 136, 105493. https://doi.org/10.1016/j.jmbbm.2022.105493
  2. O'Connor, R. A., Cahill, P. A., & McGuinness, G. B. (2021). Effect of electrospinning parameters on the mechanical and morphological characteristics of small diameter PCL tissue engineered blood vessel scaffolds having distinct micro and nano fibre populations–A DOE approach. Polymer Testing, 96, 107119. https://doi.org/10.1016/j.polymertesting.2021.107119
  3. Dolgin, J., Hanumantharao, S. N., Farias, S., Simon Jr, C. G., & Rao, S. (2023). Mechanical properties and morphological alterations in fiber-based scaffolds affecting tissue engineering outcomes. Fibers, 11(5), 39. https://doi.org/10.3390/fib11050039
  4. Alharbi, N., Daraei, A., Lee, H., & Guthold, M. (2023). The effect of molecular weight and fiber diameter on the mechanical properties of single, electrospun PCL nanofibers. Materials Today Communications, 35, 105773. https://doi.org/10.1016/j.mtcomm.2023.105773
  5. Rashid, T. U., Gorga, R. E., & Krause, W. E. (2021). Mechanical properties of electrospun fibers—a critical review. Advanced Engineering Materials, 23(9), 2100153. https://doi.org/10.1002/adem.202100153

Hydrolytic degradation section

Between lines 389 to 393: Recently Cole et al. 2020, studied the mechanical and degradation properties of PMMA and borate BG (BBG) cements through different in vitro tests. They found that the addition of BBG in different concentrations is able to maintain the mechanical properties and controlled ion release for at least 21 days of study, which could improve the in vivo osteoconductive capacity of the material [60].

  1. Cole, K. A., Funk, G. A., Rahaman, M. N., & McIff, T. E. (2020). Mechanical and degradation properties of poly(methyl methacrylate) cement/borate bioactive glass composites. Journal of Biomedical Materials Research Part B: Applied Biomaterials. https://doi.org/10.1002/jbm.b.34606

In vitro cell viability section

Between lines 415 to 418: Xing et al. 2013, studied the response of osteoblasts in electrospun PMMA fibers bio-reinforced with HA. They found that these nanocomposites improved cell organization, increased the ALP activity and accelerated osteoblast differentiation compared to pure PMMA fibers [65].

  1. Xing, Z. C., Han, S. J., Shin, Y. S., Koo, T. H., Moon, S., Jeong, Y., & Kang, I. K. (2013). Enhanced osteoblast responses to poly (methyl methacrylate)/hydroxyapatite electrospun nanocomposites for bone tissue engineering. Journal of Biomaterials Science, Polymer Edition, 24(1), 61-76. https://doi.org/10.1163/156856212x623526

Between lines 427 to 433: Recently, Zaszczyńska et al. 2024 developed PMMA and n-HA electrospun fibers in different fiber orientations. They studied the in vitro biocompatibility of the scaffolds with human osteoblastic cells of the MG63 cell line. Their results indicated that all scaffolds including those based on pure PMMA do not exhibit cytotoxicity and increase cell viability. In addition, they concluded that the addition of nanoparticles improves cell-scaffold interaction, promoting better cell adhesion in the electrospun fibers [66].

  1. Zaszczyńska, A., Kołbuk, D., Gradys, A., & Sajkiewicz, P. (2024). Development of Poly (methyl methacrylate)/nano-hydroxyapatite (PMMA/nHA) Nanofibers for Tissue Engineering Re-generation Using an Electrospinning Technique. Polymers, 16(4), 531. https://doi.org/10.3390%2Fpolym16040531

In vivo biocompatibility section:

Between lines 457 to 463: In addition, Cui et al. 2017 reported by in vitro and in vivo analyses the improved osteointegration ability of PMMA-based bone cements with the addition of Sr-BBG. For the in vivo assays, scaffolds were implanted into the medial tibial metaphysis of Sprague-Dawley rat tibiae for 8 and 12 weeks. The researchers found that Sr-BBG in PMMA stimulated new bone formation around the interface with the host bone at 8 and 12 weeks post-implantation, whereas PMMA only promoted the development of an intermediate layer of connective tissue [68].

  1. Cui, X., Huang, C., Zhang, M., Ruan, C., Peng, S., Li, L., ... & Pan, H. (2017). Enhanced osteointegration of poly (methylmethacrylate) bone cements by incorporating strontium-containing borate bioactive glass. Journal of the Royal Society Interface, 14(131), 20161057. https://doi.org/10.1098%2Frsif.2016.1057

Finally, reference 1 has been replaced:

Mistry, A.S.; Mikos, A.G. Tissue engineering strategies for bone regeneration, Adv Biochem Eng Biotechnol. 94 (2005) 1–22. https://doi.org/10.1007/b99997

For a more up-to-date reference:

  1. Jang, J. W., Min, K. E., Kim, C., Shin, J., Lee, J., & Yi, S. (2023). Correction: Review: Scaffold Characteristics, Fabrication Methods, and Biomaterials for the Bone Tissue Engineering. Interna-tional Journal of Precision Engineering and Manufacturing, 24(5), 887-887. https://doi.org/10.1007/s12541-022-00755-7

3.The short names of several groups such as PLA/PMMA (50/50) or PLA/N-BG 10 wt% etc. should be changed to short abbreviations, similarly, the very long sentences in the introduction and results section should be revised for more clarity and explanation.

Response: Thanks for your suggestion, to improve the redaction form, and to be clearer on the discussion in this manuscript, the table 3.1 was incorporated where each sample was renamed as a code form and also are detailed the quantities and ratio of each sample.

The next sentences were incorporated in the lines 514 to 516.

The prepared samples will be specified using simpler codes, to facilitate their understanding, these codes and the correct quantities and ratios are detailed in table 3.1.

Finally, the new code for each sample were incorporated into all figures on the manuscript.

Table 3.1. Amount, ratio and codes used in the preparation of PLA/PMMA electrospun fibers

Sample

Code

Amount PLA [g]

Amount PMMA [g]

Amount n-BG [g]

Neat PLA

PP-01

1.250

0

0

PLA/PMMA (75/25)

PP-02

0.9375

0.3125

0

PLA/PMMA (50/50)

PP-03

0.6250

0.6250

0

PLA/PMMA (25/75)

PP-04

0.3125

0.9375

0

Neat PMMA

PP-05

0

1.250

0

PLA + 10 wt.% n-BG

PPBG-01

1.125

0

0.1250

PLA/PMMA (75/25) + 10 wt.% n-BG

PPBG-02

0.8437

0.2813

0.1250

PLA/PMMA (50/50) + 10 wt.% n-BG

PPBG-03

0.5625

0.5625

0.1250

PLA/PMMA (25/75) + 10 wt.% n-BG

PPBG-04

0.2813

0.8437

0.1250

PMMA + 10 wt.% n-BG

PPBG-05

0

1.125

0.1250

  1. In Figure 2.8, the graph for cell viability can be changed to single graph for day 1, 3 and 7, rather than three separate graphs. Secondly, please mention the procedure, how cell viability (%) was calculated. In figure 2.8 legend DNEM should be replaced to DMEM.

Response: Thanks for your recommendation, effectively we take your proposal and modified the figure 2.8 and were incorporated follow your suggestion.

The new legend of figure 2.8 was incorporated in the lines 436-439.

Figure 2.8. In vitro cell viability of scaffolds after 1,3 and 7 days of cell culture by colorimetric assay using the MTT reagent in DMEM medium (n=3). (A) statistical analysis of cell viability using ANNOVA analysis and Bonferroni postreatment for significance, and (B) Fluorescence Microscopy using a DAPI as a cell nucleus marker.

  1. Please mention the limitations of the current study and its future implications in the discussion section.

Response: Thanks for your comments, we believe that a possible limitation of our work is mainly associated with the non-degradability of the PMMA, which would imply a considerable disadvantage in bone regeneration therapies, which require the material to degrade as the new tissue is formed. However, it is possible to project this material to other therapies that do require that the material be durable over time. Considering this and following your suggestion, we have incorporated the following sentence into the text in the hydrolytic degradation section in the lines 394 to 399.

Finally, considering the results, it is important to highlight that the non-degradability of PMMA may be a limitation when considering bone regeneration therapies, because it would imply a second operation to remove the implant. However, non-degradable polymers are widely used in other therapies as orthopedic substitutes for the treatment of bones, cartilage, hips as well as dental applications, standing out for their structural stability as well as their mechanical resistance to wear. [23,62].

  1. Ramanathan, S., Lin, Y. C., Thirumurugan, S., Hu, C. C., Duann, Y. F., & Chung, R. J. (2024). Poly (methyl methacrylate) in Orthopedics: Strategies, Challenges, and Prospects in Bone Tissue Engineering. Polymers, 16(3), 367. https://doi.org/10.3390/polym16030367
  2. Chelu, M.; Musuc, A.M. Advanced Biomedical Applications of Multifunctional Natural and Synthetic Biomaterials. Processes202311, 2696. https://doi.org/10.3390/pr11092696
  3. It is suggested to completely write the text for every figure first, then add the figure at last rather than describing the results after figure legend etc.  

Response: Thanks, according to your comments, all the figures in the manuscript was added after the discussion text.

Round 2

Reviewer 2 Report

Comments and Suggestions for Authors

This reviewer would like to sincerely thank the authors for their careful review as well as clear indication of the revisions in the revised manuscript.